# Regulation of Gene Expression of phiEco32-like Bacteriophage 7-11

**DOI:** 10.3390/v14030555

**Published:** 2022-03-08

**Authors:** Daria Lavysh, Vladimir Mekler, Evgeny Klimuk, Konstantin Severinov

**Affiliations:** 1Institute of Molecular Genetics of National Research Centre «Kurchatov Institute», 123182 Moscow, Russia; jonikl@gmail.com; 2Waksman Institute for Microbiology and Department of Molecular Biology and Biochemistry, Rutgers, State University of New Jersey, Piscataway, NJ 08854, USA; mekler@waksman.rutgers.edu; 3Center for Life Sciences, Skolkovo Institute of Science and Technology, 121205 Skolkovo, Russia

**Keywords:** bacteriophage, alternative sigma factor, transcription regulation

## Abstract

*Salmonella enterica* serovar Newport bacteriophage 7-11 shares 41 homologous ORFs with *Escherichia coli* phage phiEco32, and both phages encode a protein similar to bacterial RNA polymerase promoter specificity σ subunit. Here, we investigated the temporal pattern of 7-11 gene expression during infection and compared it to the previously determined transcription strategy of phiEco32. Using primer extension and in vitro transcription assays, we identified eight promoters recognized by host RNA polymerase holoenzyme containing 7-11 σ subunit SaPh711_gp47. These promoters are characterized by a bipartite consensus, GTAAtg-(16)-aCTA, and are located upstream of late phage genes. While dissimilar from single-element middle and late promoters of phiEco32 recognized by holoenzymes formed by the phi32_gp36 σ factor, the 7-11 late promoters are located at genome positions similar to those of phiEco32 middle and late promoters. Two early 7-11 promoters are recognized by the RNA polymerase holoenzyme containing the host primary σ^70^ factor. Unlike the case of phiEco32, no shut-off of σ^70^-dependent transcription is observed during 7-11 infection and there are no middle promoters. These differences can be explained by the fact that phage 7-11 does not encode a homologue of phi32_gp79, an inhibitor of host and early phage transcription and an activator of transcription by the phi32_gp36-holoenzyme.

## 1. Introduction

*Salmonella enterica* serovar Newport phage 7-11 is a podovirus with a distinctive, strongly elongated virion head [1]. Phylogenetic analysis reveals that it is related to coliphage phiEco32; however, it belongs to unclassified *Podoviridae*. The *E. coli* phage phiEco32 was considered to be unique when its genomic sequence was first determined [2]. However, subsequently, multiple related phages were isolated in Canada, China, South Korea and elsewhere [3,4,5]. PhiEco32 remains the best-studied member of the *Kuravirus* genus to date. As most bacteriophages, it relies on host RNA polymerase (RNAP) for its development within the host cell [6,7,8]. Early phage genes are transcribed from four promoters located at the beginning of the early gene cluster. These promoters are recognized by the housekeeping σ^70^ RNAP holoenzyme of the host. All late genes and at least some of the middle genes of phiEco32 are transcribed from promoters recognized by an RNAP holoenzyme containing phage σ factor phi32_gp36. Six phi32_gp36-dependent middle and late promoters were identified experimentally. A small phage polypeptide, phi32_gp79, a product of a middle gene, inhibits transcription by the σ^70^ holoenzyme and stimulates phi32_gp36-dependent transcription in vitro [2,6]. Thus, phi32_gp79 may be responsible for regulated expression of phage genes and host transcription shut-off.

In this report, we describe the use of bioinformatics and biochemical approaches to characterize the temporal patterns of phage and host gene expression during the infection by phage 7-11, a distant relative of phiEco32, and compare and contrast its transcription strategy to phiEco32 and related phages.

## 2. Materials and Methods

### 2.1. Comparison of phiEco32-like Genomes and Prediction of Their Promoters

The comparison of phiEco32-like bacteriophage ORFs was conducted using BLASTP. The σ^70^-promoters in phage 7-11 genome were searched for as described previously [6]. PhiEco32_gp36 promoter and SaPh711_gp47 promoter recognition profiles were constructed using SignalX [9]. Search of candidate promoters in the phage genome was conducted using the GenomeExplorer software package [9].

### 2.2. Bacterial Strains, Phage and Growth Conditions

*E. coli* XL10-Gold (Δ (mcrA)183 Δ(mcrCB-hsdSMR-mrr)173 endA1 supE44 thi-1 recA1 gyrA96 relA1 lac Hte [F′ proAB lacIq ZΔM15 Tn10 (Tetr) Amy Camr]) ultracompetent cells (Stratagene) were used for molecular cloning [10].

*E. coli* B *BL21(DE3) (F-dcm ompT hsdS(rB-mB-) gal λ(DE3))* (Stratagene) was used for recombinant proteins’ overproduction.

All bacterial strains were grown in LB media (1% Bactotryptone, 1% NaCl and 0.5% yeast extract, with or without 1.5% Bactoagar) at 37 °C with appropriate antibiotics.

Bacteriophage 7-11 and its host *S. enterica* serovar Newport were obtained from the Félix d’Hérelle Reference Center for Bacterial Viruses (Université Laval, Quebec, Canada). Bacterial cultures were grown and phage infection was performed in liquid LB media at 37 °C with orbital shaking. To prepare 7-11 lysates, one plaque and 100 µL of overnight culture of *S.* Newport were added to 100 mL LB media and incubated with shaking at 37 °C for 4 h. The culture was treated with 1% chloroform and centrifuged for 30 min at 5000× *g*. The titer of resulting phage lysate usually was ~4 × 10^10^ p.f.u./mL. Phage DNA was purified from 5 mL of lysate using Thermo Fisher Scientific column DNA extraction kit. For RNA purification, cells were grown to an OD_595_ = 0.4 and infected with 7-11 phage at MOI (multiplicity of infection) of 10. At this multiplicity, the efficiency of infection of the host was above 80%. Infection was stopped at various time points (including a zero-point, which corresponds to uninfected cells) by rapid chilling and by the addition of rifampicin to a final concentration of 30 µg/mL. 15 mL aliquots of infected cultures were collected for each time point and cells were collected by centrifugation at 5000× *g* for 15 min and used for total RNA purification.

### 2.3. RNA Purification

Total RNA was isolated by two extractions with hot phenol (pH 4.5) followed by ethanol precipitation as previously described [8,11]. After the extraction, RNA was treated with 15 u of RNase-free DNase I (ThermoFisher Scientific) at 37 °C for 30 min. Reactions were terminated by the addition of EDTA to 5 mM and by heating at 65 °C for 10 min. Samples were phenol-extracted and ethanol-precipitated. RNA was dissolved in DEPC-treated water and stored at −70 °C.

### 2.4. Primer Extension Analysis

Primers were 5′-end-labeled with (γ-^32^P)-ATP using phage T4 PNK (New England Biolabs) as recommended by the manufacturer. Annealing of 1 pmol of a mixture of (γ-^32^P)-labeled primers with 10 μg of total RNA was performed in 40 mM PIPES (pH 6.4), 400 mM NaCl, 1 mM EDTA and 80% formamide. Samples were denatured at 85 °C for 10 min and then cooled on ice. The annealing continued overnight at 0 °C. Next, samples were precipitated with ethanol, washed with 70% and 96% ethanol, dried and dissolved in DEPC-treated water. Primers were extended by Maxima Reverse Transcriptase (Fermentas) according to manufacturer’s recommendations. Equal volume of formamide loading buffer (80% formamide, 10 mM EDTA, 0.05% bromphenol blue and 0.05% xylene cyanol) was added to the samples; after which, they were heated for 5 min at 85 °C and cooled on ice. For primer extension reactions using in-vitro-synthesized RNA as templates, RNA was produced by in vitro transcription from appropriate PCR fragments by the *E. coli* core RNAP supplemented with recombinant SaPh711_gp47 in the presence of 100 μM of each NTP at conditions described in the in vitro transcription section. RNA was cleaned up with QIAGEN RNeasy Mini Kit and used for primer extension. Sequence marker lanes were prepared by setting up sequencing reactions with the USB Thermo Sequenase Cycle Sequencing Kit on PCR-amplified phage DNA fragments. Sequencing reactions were carried out with same primers as those used for primer extension reactions, thus allowing mapping of primer extension products with single-nucleotide resolution. The reaction products were separated on 6 or 8% denaturing polyacrylamide sequencing gels; following this, autoradiography and visualization using PhosphorImager were performed.

### 2.5. 7-11 Gene 47 Cloning and SaPh711_gp47 Purification

Phage 7-11 gene *47* was cloned in between NdeI or BamHI sites of the pET18b vector. His-tagged *E. coli* RNAP core and recombinant σ^70^ subunit were prepared as previously described [8,12,13,14,15]. Phage phiEco32 proteins phi32_gp79 and phi32_gp36 as well as 7-11 protein SaPh711_gp47 were prepared as described [2].

### 2.6. In Vitro Transcription

Transcription reactions were performed in transcription buffer (40 mM Tris-HCl pH 7.9, 40 mM KCl, 10 mM MgCl_2,_ 5 mM DTT and 100 µg/mL BSA) and contained 150 nM of *E. coli* RNAP core enzyme, 450 nM of recombinant σ^70^ or 450 nM of recombinant SaPh711_gp47 protein and, if mentioned, 450 nM of recombinant phi32_gp79 and 20 nM of DNA template. Reactions were incubated for 10 min at 37 °C, followed by the addition of 100 µM ATP, CTP and GTP; 10 µM UTP and 0.4 µCi of [α-^32^P]UTP. Reactions proceeded for 10 more minutes at 37 °C and were terminated by the addition of an equal volume of formamide loading buffer. Primers used to generate templates for transcription are listed in Appendix A.

### 2.7. KmnO_4_ Probing

Promoter complexes were formed as described in the in vitro transcription section with the following modifications: 100 nM of (γ-^32^P)-5′-end-labeled DNA fragment was used and transcription buffer did not contain DTT. After preincubation for 15 min at 37 °C, the reactions were probed with 2 mM KmnO_4_ for 20 s at 37 °C. Reactions were terminated by the addition of 0.5 volume of stop solution containing 600 mM β-mercaptoethanol, followed by ethanol precipitation and 10% piperidine treatment at 95 °C for 15 min. Samples were chloroform-extracted and ethanol-precipitated. Pellets were dissolved in formamide loading buffer and reaction products were analyzed on sequencing gels.

### 2.8. Fluorometric Measurements

Fluorescence measurements were carried out at 25 °C using a QuantaMaster QM4 spectrofluorometer (PTI) in transcription buffer containing 0.02% Tween 20 [16]. The σ^70^ protein labeled at position 517 with fluorescein (σ^70^*) was obtained as previously described [17]. Final assay mixtures (800 ul) contained 1.5 nM σ^70^*, 1 or 3 nM RNAP core complexed with Rifampicin (Rif) and 12 nM gp47 (if indicated). The RNAP core–Rif complex was prepared by incubation of RNAP core (1 or 3 nM) and 1 uM Rif for 10 min at 25 °C [18]. The fluorescein fluorescence intensities were recorded with an excitation wavelength of 498 nm and an emission wavelength of 520 nm. Time-dependent fluorescence changes were monitored using manual mixing; the mixing time was 15 s.

## 3. Results

### 3.1. The phiEco32-like Phages and Their Genomes

The genome of *E. coli* phage phiEco32 was determined in 2008 [2]. At the time, phiEco32 appeared to be a novel phage, i.e., no more than 40% of the products of its ORFs had homologues in public databases [2]. Subsequently, over one hundred bacteriophages with similarities in genomic sequence and gene organization were submitted to public databases, including, among others, *E. coli* (APEC) phages NJ01 (NC_018835), ECB2 (NC_018859), SU10 (KM044272) and 172-1 (KP308307) [3,4,5]. BLASTP searches identified 69 genes common to phiEco32 and these *E. coli* phages (Appendix A). These genes thus define core genes [19] within the *Podoviridae* family *Kuravirus* group that we will refer as “the phiEco32-subgroup”.

*Salmonella enterica* serovar Newport phage 7-11 (NC_015938) and *Cronobacter sakazakii* phage vB_CsaP_GAP52 (GAP52) (NC_019402) are also related to phiEco32-like phages. Comparison of phiEco32 and 7-11 genomes is schematically presented in Figure 1. 27.15% of 7-11 and 32.03% of phiEco32 ORFs are homologous to each other [1]. The genome of phage GAP52 [20] encodes 79 proteins homologous to 7-11 gene products, which corresponds to ~52% of 7-11 ORFs and ~69% of GAP52 ORFs (Appendix A). Despite the relatively high similarity of many proteins, the genomes of GAP52 and 7-11 share only 10% similarity at the nucleotide sequence level. Among the proteins common to 7-11 and GAP52, only 33 share common descent with phiEco32 proteins. Surprisingly, only 23 of these correspond to phiEco32-like phages’ core gene products. Since the number of homologous proteins between phiEco32 and 7-11/GAP52 is less than that shared by other *Kuravirus* phages, we consider phages 7-11 and GAP52 as members of a distinct “7-11 subgroup” among the phiEco32-like bacteriophages.

Transcription of phiEco32 genome was studied by macroarray and primer extension analysis and conventional early, middle and late expressed classes of phage genes were revealed [6]. Since phiEco32-like bacteriophages share a common overall genome architecture, one can assume that genes of other phiEco32-like phages can be similarly divided into three expression classes. In the schematic representation of phiEco32 and 7-11 genomes (Figure 1), extended clusters of genes that encode structural virion proteins and DNA packaging machinery components can be identified. These genes (colored red in Figure 1 and shown to belong to the late expression class in phiEco32) are transcribed in the rightward direction and occupy about one-third of each genome. The second cluster (colored green in Figure 1 and shown to belong to the middle expression class in phiEco32) includes shared genes coding for phage DNA polymerase, nucleotide metabolism enzymes and a host RNA polymerase σ subunit. Finally, the cluster of genes colored blue in Figure 1 (shown to belong to the early expression class in phiEco32) is most diverse between the two phages and contains mostly genes of unknown functions that likely play a role in host takeover [21]. The middle and early genes are transcribed in the direction opposite to that of late gene cluster transcription.

All phiEco32-like phages encode a σ factor (phi32_gp36 in phiEco32, a product of a middle gene). Previously, we identified six phi32_gp36 promoters located upstream of middle and late genes of the phage [6]. These promoters are characterized by a single highly conserved consensus sequence, tAATGTAtA. Transcription start sites are located 6–8 bp downstream of the phi32_gp36 holoenzyme promoter element. We searched for phi32_gp36-like promoters in every phiEco32-like phage. Sequences similar to phiEco32 middle/late promoter consensus elements were found in expected locations for every member of the phiEco32-subgroup phages (Appendix A). However, no matching sequences were found in the 7-11 and GAP52 genomes.

The putative σ factor of 7-11, SaPh711_gp47, is a distant homologue (15% identity) of phi32_gp36 and other phiEco32 subgroup σ factors, which likely explains our inability to find 7-11 promoters using the phiEco32 middle and late promoter consensus motifs. We have searched all 7-11 intergenic regions (that were defined as DNA sequences between nucleotides positions “−100” and “+200” relative to predicted start codons) for the presence of overrepresented DNA motifs by MEME suite [22] and the Align2N algorithm. The intergenic regions were also scanned for motifs similar to those of promoter consensus elements of known bacterial promoters using the Prodoric algorithm [23]. No overrepresented motifs or putative promoters were identified using this procedure.

Transcription factor phi32_gp79 inhibits σ^70^-dependent hosts and early phage phiEco32 transcription but activates phi32_gp36-dependent transcription in vitro [6]. Phi32_gp79 homologues can be predicted in all phiEco32-like phages. No phi32_gp79 homologues are encoded by either the 7-11 or GAP52 genomes.

### 3.2. Identification of 7-11 Late Promoters

One can expect that there shall be (at least) two types of promoters in the 7-11 genome: early promoters dependent on the host RNAP holoenzyme containing the primary σ^70^ factor and middle and/or late promoters dependent on the holoenzyme containing SaPh711_gp47. Since we could not predict SaPh711_gp47 promoters bioinformatically, we decided to proceed from the fact that out of six phiEco32 genes (*6, 13, 26, 40, 58* and *68*) located downstream of phi32_gp36-dependent promoters (in Figure 1, shown by dark shade or green (for middle) and red (late) genes), five are present in the 7-11 genome. We reasoned that noncoding regions directly upstream of these genes or in front of the first genes of operons that contain these genes may house SaPh711_gp47-RNAP holoenzyme promoters. In order to confirm this hypothesis, total RNA was purified from 7-11 infected *S. enterica* serovar Newport cultures at different time points postinfection and analyzed by primer extension, with primers annealing downstream of possible SaPh711_gp47-dependent promoters. Distinct primer extension products were detected in reactions with four primers annealing to intergenic regions of genes *1*, *8*, *16* and *48* in RNA samples prepared from cells collected 40 min postinfection (Figure 2A). Since the infection cycle of 7-11 is ca. 60 min, late promoters were expected to be active at the chosen time point.

The products of primer extension shown in Figure 2A were loaded on sequencing gels alongside marker sequencing reactions performed with the same primers using PCR fragments carrying intergenic regions under study as templates. Alignment of sequences upstream of mapped primer extension products’ 5′ ends is shown in Figure 2B. As can be seen, a bipartite GTAAtg -(16)- aCTA consensus motif is present in all four sequences.

We next used the SignalX program [9] to create a pattern describing the consensus element based on four experimentally identified sequences and searched the 7-11 genome with this pattern. The search retrieved four additional matching sequences upstream of genes *12*, *22, 28* and *69* (Figure 2C). Primer extension analysis revealed the presence of expected products for primers annealing downstream of genes *22, 28* and *69* (Figure 2D).

Time-resolved multiplex analysis showed that all primer extension products studied above became visible 20 min postinfection and reached a maximum at 50 min postinfection, and then their abundance decreased, likely due to cell lysis that became apparent 60 min postinfection (Figure 2E). Thus, the kinetics of accumulation of primer extension products corresponded to the late expression class. We therefore conclude that the bipartite motif, whose logo is presented in Figure 2D, defines the consensus of late promoters of phage 7-11.

### 3.3. SaPh711_gp47 as a Late Sigma Factor

To determine whether SaPh711_gp47 is responsible for late transcription of bacteriophage 7-11, the SaPh711_gp47 gene was cloned into an *E. coli* expression vector. Recombinant protein was purified and tested for its ability for direct transcription by the *E. coli* RNAP core (99% amino acid sequence identity to the *S. enterica* enzyme) from DNA fragments that contained identified 7-11 late promoter sequences. Efficient in vitro RNA synthesis was observed from each late promoter detected in vivo (Figure 3A). In addition, the P12, which showed no activity in vivo, was active in in vitro transcription (Figure 3A). The sizes of transcripts detected by primer extension with gene-specific primers using RNA purified from infected cells matched those detected with in-vitro-transcribed RNA (data shown for P_48_ in Figure 2A).

As expected, the RNAP holoenzyme containing the σ^70^ subunit did not produce transcripts from DNA fragments containing late phage promoters in vitro (Figure 3B, data shown for P_1_). Conversely, the SaPh711_gp47-holoenzyme did not transcribe from a DNA template containing a strong σ^70^-promoter T7 A1 [24]. The addition of SaPh711_gp47 to reactions containing the σ^70^ RNAP holoenzyme did not lead to the appearance of bands corresponding to late phage promoter transcripts, indicating that the two σ factors compete for the same RNAP core binding site. Surprisingly, the dual transcription regulator of phi32_gp79 that inhibits σ^70^ transcription in vitro and activates phi32_gp36-dependent transcription [6] inhibited SaPh711_gp47 transcription in vitro (Figure 3B).

Promoter melting at several late 7-11 promoters was analyzed by potassium permanganate (KMnO_4_) probing. Since thymines in single-stranded, but not double-stranded, DNA are sensitive to oxidation by KMnO_4_, this reagent allows one to determine the size of the transcription bubble in the promoter complex. As can be seen, the region of localized DNA melting included the TSS and the -10 promoter element (Figure 3C) [25]. Thus, SaPh711_gp47 is a σ factor that directs host RNAP core to transcribe late 7-11 promoters located in front of genes *1*, *8*, *12*, *16*, *22*, *28*, *48* and *69*.

### 3.4. SaPh711_gp47 Sigma Factor Forms a Stable Complex with Host Core RNAP

The affinity of SaPh711_gp47 to core RNAP was analyzed in competition experiments with σ^70^ subunit fluorescently labeled at amino acid 517 (σ^70^*) (Figure 4).

In the context of the holoenzyme, the labeled amino acid of σ^70^* is adjacent to the binding site of rifampicin (Rif), an inhibitor of transcription elongation that interacts with RNAP core [26]. Upon formation of the holoenzyme in the presence of Rif, the fluorescence of σ^70^* decreases, due to quenching of the σ^70^* fluorophore by Rif via the FRET mechanism (black curve) [17]. The degree of fluorescence quenching depends on the ratio of core and σ^70^ (red curve). When RNAP core bound to rifampicin was preincubated with SaPh711_gp47, followed by the addition of fluorescently labeled σ^70^*, no quenching was observed, even after a 1 h incubation with σ^70^* (blue curve), presumably because the σ^70^ binding site was occupied by SaPh711_gp47. When σ^70^ and SaPh711_gp47 were added to the Rif-bound RNAP core simultaneously, fluorescence quenching (green curve) was only moderately less than that observed in the same ratio of RNAP core/σ^70^ in the absence of SaPh711_gp47 (red curve). Since this experiment was performed under conditions when the concentration of SaPh711_gp47 was eight times greater than that of σ^70^, it follows that the formation of the holoenzyme with σ^70^ is faster than the formation of the SaPh711_gp47 holoenzyme. However, if the complex with SaPh711_gp47 is already formed, σ^70^ is unable to displace bound SaPh711_gp47.

### 3.5. Identification of 7-11 σ^70^ -Dependent Promoters

Five putative σ^70^-depedent promoters were predicted in the phage 7-11 genome (Figure 5A) [1]. In order to confirm this prediction and monitor the behavior of early transcripts during the infection, primer extension reactions with primers annealing downstream of predicted σ^70^-dependent promoters were performed on total RNA purified from 7-11 infected cultures collected at different time points postinfection. Out of five predicted early promoters, only two were validated in vivo by primer extension (Figure 5B). These promoters were located upstream of gene *151*, at the very right end of the genome. Figure 3B also shows changes in abundance of primer extension products for viral RNA transcribed from these promoters as well as for a σ^70^-dependent host *ompX* transcript throughout the infection. Primer extension products corresponding to viral σ^70^-dependent promoters appeared 5 min postinfection and their abundance remained constant throughout the infection. Likewise, the amount of primer extension product corresponding to host *ompX* gene promoter remained constant throughout the infection. It thus follows that there is no mechanism for σ^70^-dependent transcription shut-off during phage 7-11 infection.

For phage phiEco32, a bioinformatic search resulted in prediction of a putative σ^70^-dependent promoter within gene *37*, which could be used for transcription of downstream phiEco32 sigma factor gene *36* [6]. For phage 7-11, we and others failed to predict a σ^70^-promoter in the intergenic region between genes *47* and *48* or within gene *48*. Nevertheless, we performed primer extension reactions using RNA from infected cells and a primer annealing downstream of gene *47*. A distinct primer extension product was detected; however, its accumulation pattern corresponded to late transcript class (Figure 5B). No SaPh711_gp47 promoter consensus was located upstream of the mapped position of primer extension end point, which mapped in the intergenic region between genes *48* and *47*. This intergenic region contained a sequence that could form a stem-loop structure with a GTTCGG loop (Figure 5C). The stem-loop structure was also found in the corresponding position of phage GAP52 genome. We assumed that the primer extension product that mapped upstream of the 7-11 σ gene arose not due to transcription initiation but was either a result of in vivo processing or in vitro reverse transcription stalling on a hairpin formed on a phage transcript.

## 4. Discussion

In this work, we report analysis of temporal gene expression regulation of phiEco32-like bacteriophage 7-11. Unlike phiEco32 and most other lytic phages, 7-11 genes belong to just two expression classes: early and late. Early transcription initiates from two strong σ^70^-dependent promoters located at the right end of the genome and proceeds leftward, resulting in long polycistronic transcripts covering almost half of the genome. The first late promoter is located upstream of gene *69.* Gene *47*, coding for the 7-11 σ factor responsible for late transcription, is presumably transcribed, at least initially, by RNAP molecules that initiate transcription from early σ^70^-dependent promoters. Since *47* is located deep in the leftward transcribed cluster of late phage genes, there should be some transcription read-through from early promoters to allow late transcription to occur. It is highly likely that 7-11 employs some strategy to antiterminate transcription that remains to be defined. The amount of gene *47* transcripts can subsequently strongly increase through a positive feedback loop due to activity of late P_48_ promoter, thus orchestrating a switch from early to late phage transcription.

Robust production of SaPh711_gp47 in amounts necessary to compete with σ^70^ may be further stimulated by a sequence in the intergenic region between genes *47* and *48* that can form a stem-loop structure with a GTTCGG loop (Figure 5C). An identical sequence is found in the corresponding place of GAP45, a 7-11 relative that has only 10% identity with 7-11 genome-wide. Similar sequences (X)_2-8_CUUCGG(Y)_2-8_ (where X is complement to Y, G-U pairs allowed) are commonly found, predominantly in intergenic regions, of many phages [27]. They are especially abundant in T4 phages and it has been shown that primer extension reactions by reverse transcriptase terminate at T4 CUUCGG hairpins [27]. The known functions of such hairpins vary. In T4, a 5′ CUUCGG hairpin confers stability to the T4 gene *32* (DNA polymerase) transcript [28,29]. The highly structured, single-stranded RNA genome of the MS2 phage contains a CUUCGG hairpin at its 3’ end, where it was shown to be involved in temporally regulated rounds of translation and replication of the phage genome [30]. We propose that a similar mechanism may be operational in the case of the 7-11 and GAP45 late σ transcripts.

There does not appear to be specific shut-off of either early phage or host transcription during 7-11 development. Presumably, SaPh711_gp47, which binds RNAP core better than σ^70^, is alone able to efficiently compete for the binding such that enough SaPh711_gp47 holoenzyme to serve the needs of the virus is formed. The lack of host transcription shut-off and the absence of the middle expression class of phage genes may be due to the absence of a homologue of phi32_gp79, a middle gene product that inhibits σ^70^ transcription and activates phi32_gp36-dependent transcription [6].

Promoters recognized by SaPh711_gp47 consist of two consensus elements: GTAAtg –(16)- aCTA. Promoters recognized by phi32_gp36 consist of a single consensus element tAATGTAtA. For both cases, transcription start sites are located 6–9 bp upstream relative to the promoter sequence. The absence of the -35 element in phi32_gp36 promoter can be explained by the absence of homology with region 4 of other sigma factors (region 4 is responsible for the recognition of -35 promoter elements) in phi32_gp36. Conversely, the amino acid sequence in the C-terminal part of SaPh711_gp47 is homologous to region 4 of known σ factors and should be therefore capable of sequence-specific recognition of DNA.

The 7-11 σ factor, SaPh711_gp47, is a distant homologue of phi32_gp36 and other phiEco32-like phages’ σ factors (~15% similarity). The closest homologues of SaPh711_gp47 (~50% similarity) are σ ^24^ from *Winogradskyella sp.* PG-2 (WP_045474069.1), a σ^70^ family factor from *Aeribacillus pallidus* CIC9 (WP_094245485.1) and a hypothetical protein from *Desulfuromonadales bacterium* (HIJ97285.1). Promoter specificity for RNAP holoenzymes containing these most closely related to SaPh711_gp47 proteins is unknown. Given the evolutionary distance between phiEco32 and 7-11 σ factors, it is clear that the ancestors of these phages must have acquired their σ factor genes independently and from different sources. Yet, the locations (but not the sequences) of promoters recognized by phi32_gp36 and SaPh711_gp47 holoenzymes in their respective genomes are conserved, providing an interesting evolutionary example of how functional constraints on the expression strategy of phage genes lead to common but independent solutions. Given that organized transcription of nonearly phage genes in phi32 and 7-11 group phages depends on independently acquired σ factors recognizing multiple distinct promoters, we predict that further analysis of related phages shall either reveal interesting hybrid variants or, conversely, additional phages with other, yet to be described, pairs of σ -factor genes and promoters they recognize.

## Figures and Tables

**Figure 1 viruses-14-00555-f001:**
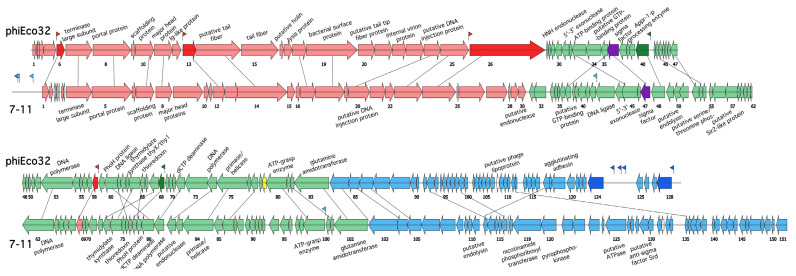
**Comparison of phiEco32 and 7-11 genomes.** The genomes of phages phiEco32 (NC_010324) and 7-11 (NC_015938) are schematically presented. Arrows represent annotated genes; the direction of an arrow indicates the direction of transcription. The ORFs are colored according to experimentally determined temporal classes of phiEco32: blue—early genes, green—middle genes and red—late genes. Dark shaded colors indicate phiEco32 genes, in front of which promoters were located experimentally (shown as appropriately colored flags). ORFs’ encoding transcription factors are colored violet (sigma factors) and yellow (inhibitor of σ^70^ transcription phi32_gp79). Homologous genes in the two genomes are connected by lines. For 7-11, bioinformatically predicted σ^70^ promoters are indicated with colors matching expected gene expression classes.

**Figure 2 viruses-14-00555-f002:**
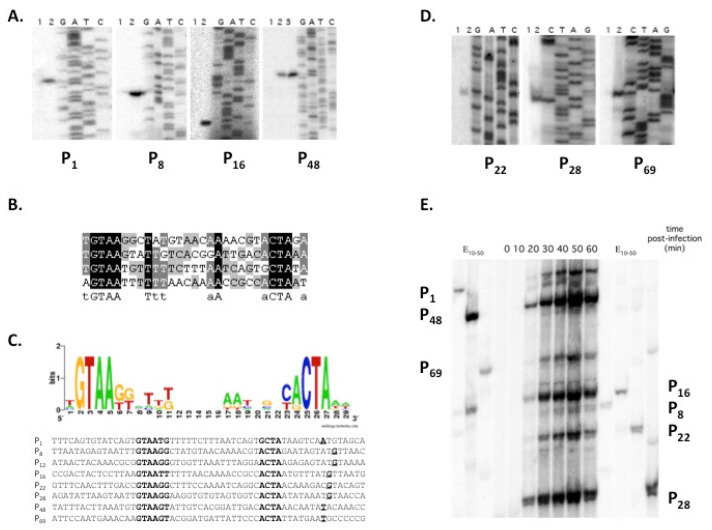
**Identification of bacteriophage 7-11 late promoters:** (**A**) Results of primer extension analysis using RNA prepared from cells before (lanes 1) or 40 min after (lanes 2) 7-11 infection using primers annealing upstream of 7-11 genes *1*, *8*, *16* and *48* corresponding to phiEco32 genes proximal to its late promoters. Lane 3 for the P_48_ panel shows the results of primer extension with RNA synthesized in vitro by the SaPh711_gp47-RNAP holoenzyme. The reaction products were separated in 6% denaturing polyacrylamide gels and visualized using a PhosphorImager. (**B**) The alignment of P_1_, P_8_, P_16_ and P_48_ promoters validated in panel (**A**). (**C**) A Logo representation of phage 7-11 late promoters. The alignment below shows, in addition to P_1_, P_8_, P_16_ and P_48_ promoters validated in panel (**A**), promoters retrieved by an additional targeted search using the 7-11 late promoter consensus. Conserved elements are shown in bold typeface. Transcription start points are shown in bold and are underlined. (**D**) Validation of P_22_, P_28_ and P_69_ promoters predicted bioinformatically in (**C**). See panel (**A**) legend for details. (**E**) Kinetics of accumulation of late transcripts during 7-11 infection. Results of multiplex primer extension analysis of total RNA extracted from phage-infected cells collected at various time points postinfection using primers designed to reveal transcription from indicated late phage promoters. On the left and on the right, products of individual primer extension reactions on RNA pooled from samples from each time point are shown. The reaction products were separated in 6% denaturing polyacrylamide gel and visualized using PhosphorImager.

**Figure 3 viruses-14-00555-f003:**
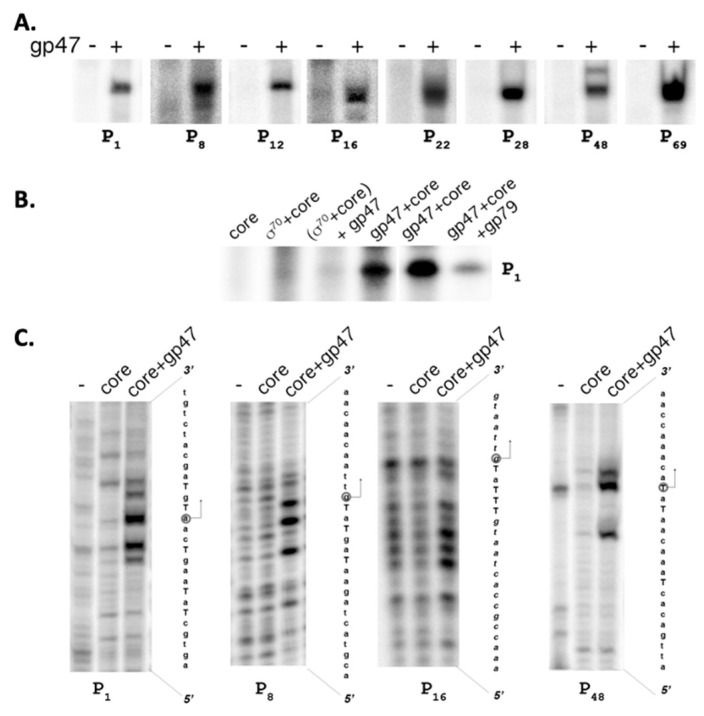
**SaPh711_gp47 is a sigma factor required for late promoter transcription in vitro**: (**A**) The products of in vitro transcription reaction by *E. coli* RNAP core enzyme in the presence or in the absence of three-fold molar excess of recombinant SaPh711_gp47 from DNA fragments corresponding to intergenic regions in front of 7-11 genes *1*, *8, 16*, *22, 28, 48 and 69* are shown. The mapping of the product of P_48_ transcription by primer extension is shown in Figure 2A. (**B**) Analysis of in vitro transcription products by *E. coli* RNAP core in the presence of σ^70^, SaPh711_gp47 or both and with the addition of phiEco32 inhibitor of σ^70^ transcription gp79 from a DNA fragment corresponding to intergenic regions in front of 7-11 gene *1*. The reaction products were separated in a 10% denaturing polyacrylamide gel and visualized using PhosphorImager. (**C**) DNA fragments harboring indicated late 7-11 promoters, terminally labeled at the template strand, were probed with KMnO_4_ in the absence or in the presence of core RNAP with or without SaPh711_gp47. The reaction products was separated by 6% denaturing PAGE and revealed using PhosphorImager. Mapping of permanganate sensitive bands to DNA sequence is shown at the right of each panel.

**Figure 4 viruses-14-00555-f004:**
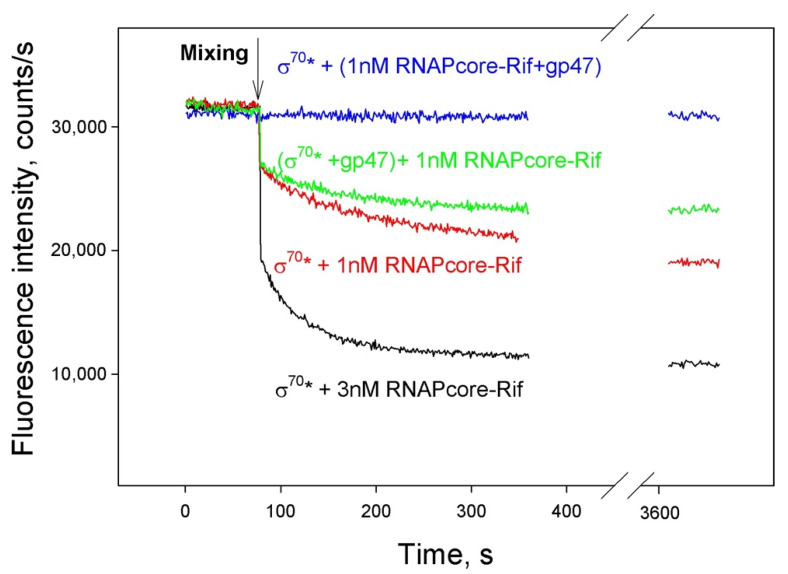
**Binding of SaPh711_gp47 to *E. coli* RNA core enzyme**. The affinity of gp47 for *E. coli* RNAP core enzyme was evaluated by a real-time fluorometric competition binding assay using a derivative of the *E. coli* RNAP σ^70^ subunit labeled with fluorescein (σ^70^*) as a reference competitor. Shown are time dependences of the decrease in fluorescence of the RNAP core–Rif complex upon the addition to σ^70^* in the absence (black and red curves) and in the presence (blue and green curves) of gp47. Black curve shows decrease in σ^70^* fluorescence upon the formation of the holoenzyme, since rifampicin is quenching the fluorophore. Red curve shows that σ^70^* fluorescence quenching depends on the ratio of core and σ^70^*. The blue curve shows the lack of decrease in σ^70^* fluorescence upon the addition of RNAP core preincubated with gp47 to σ^70^*. The green curve shows time dependence of the decrease in fluorescence upon the addition of RNAP core to premixed σ^70^* and gp47. The concentrations of σ^70^*, gp47 and RNAP core were 1.5 nM, 12 nM and 1 or 3 nM, respectively, unless otherwise indicated.

**Figure 5 viruses-14-00555-f005:**
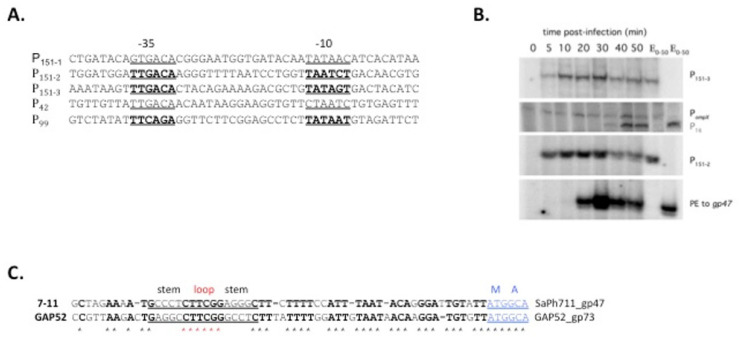
**Kinetics of accumulation of****σ^70^-dependent transcripts during 7-11 infection and identification of a potential regulatory hairpin in front of gene *47***: (**A**) Sequences of all predicted σ^70^-promoters of bacteriophage 7-11 (from NC_015938 file) are shown; putative −35 and −10 promoter consensus elements are underlined/indicated by bold typeface. (**B**) Results of multiplex primer extension analysis with total RNA extracted from phage-infected cells collected at various time points postinfection, using primers designed to reveal transcription from bioinformatically predicted σ^70^-promoters shown in panel (**A**), are shown. Only promoters in front of gene *151* (151-2 and 151-3) were validated. A primer annealing downstream of late promoter P_16_ was used as a control to indicate late gene expression. A primer annealing to *Salmonella* Newport gene *ompX* mRNA was used to follow the fate of a host transcript. Results of reaction of extension of a primer annealing downstream of gene *47* are shown at the bottom. On the right, products of extension reactions with individual primers on RNA sample prepared by mixing RNA samples from each time point (E_0–50_) are shown (a control to establish the identity of primer extension products). The reaction products were separated in 6% denaturing polyacrylamide gel and visualized using PhosphorImager. (**C**) Identification of type (X)_5_GTTCGG(Y)_5_ hairpins in the intergenic region in front of SaPh711_gp47 and phage GAP52 σ factor gp73 genes. Conserved nucleotides are shown in boldface, protein-coding sequence is shown in blue and loop of the hairpin is shown in red.

## Data Availability

There is no data that needs to be referenced here.

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
