# Peer review of "Regulation of Gene Expression of phiEco32-like Bacteriophage 7-11"

_viruses, 2022, doi:10.3390/v14030555_

Round 1

Reviewer 1 Report

Lavysh et al, 2022 review

General points.

This manuscript nicely characterises the functional genome structure, temporal regulation of transcription, sigma factor specificity and promoter elements of Salmonella enterica serovar Newport phage 7-11 genes. This phage is related to phiEco32 but differs in interesting ways in terms of sigma factor utilisation and – importantly – the host sigma70-dependent RNAP transcription is not repressed, as is the case with many (most?) bacteriophages (and archaeal viruses). Intriguingly, 7-11 employs a streamlined gene expression cascade only consisting of early and late genes. The results are compelling and clearly described, the data presented in clear fashion in the accompanying figures and the conclusions are concise and do not suffer overinterpretation.

Phage and viral gene regulation are considered to be under strong selection pressure, and I wonder about the underlying reasons for the absence of any host transcription suppression by the 7-11 factors and transcription machinery. I suggest that the authors share their hypothesis or reasoning why this should provide an selective advantage for this phage – e.g. in contrast to the closely related phiEco32 phage that they compare the sigma factors to.

Specific points.

I recommend to replace the imperative phrase ‘shall be transcribed’ with ‘is transcribed’ in lines 423 and 425.

Author Response

Dear Reviewer 1,

thank you for positive assessment of our work. We attended to your comment and changed "shall be transcribed" (which was not an imperative but rather indicated a presumption) to "is presumably transcribed".

Sincerely,

Konstantin Severinov

Reviewer 2 Report

This manuscript describes a study of patterns of gene expression in phage 7-11 over its cycle of infection, defining early and late promoters and the transcription factors involved in their regulation.  Differences in comparison to related phage phiEco32 appear to account for the lack of middle promoters and inhibition of host transcription in 7-11.

Specific comments:

(1) Line 32: Outside or inside the same genus (Phieco32virus)?

(2) Line 133: Would replace the formula with "Permanganate," or that and have the formula in parentheses.

(3) Line 155: What is the "mixing dead time?"  Please explain for those of us who don't know.

(4) Line 174: I think "at the level of DNA" would be better replaced with "at the nucleotide sequence level."

(5) Line 176: I think it is the number of homologs rather than the "amount" of them.  And are these homologs based on nucleotide sequence similarity or amino acid similarity?

(6) Lines 179-182: The intent of these two sentences is not entirely clear; please restructure them for clarity.

(7) Line 22: Add a bit of detail to "distant homolog."  What is the similarity level?

(8) Figure 1 legend: Would it be appropriate to include that the color-coded gene classifications are typical temporal pattern, since in 7-11 you ended up observing no middle expression?

(9) Line 218: member rather than "members,"

(10) Line 220: how distant are the homologs?

(11) Line 227: motifs rather than "motives."

(12) Line 236: I am confused by the use of "There shall..." in this statement; perhaps instead "We expect..." or "We pedict..."?   

(13) For line 253, the markers sequenced should be included in the detail in the Material & Methods for primer extension.  Also, insert "the" in this sentence before "same."

(14) Line 258: A comma after "sequences" would help clarity.

(15) Figure 2: Is it possible to make the test slightly larger in panel A so that the subscripts are not microscopic?  Also, in the legend insert "a" before "PhosphorImager."

(16) Line 297: Is the 99% identity at the amina acid level?

(17) Line 299-300: Can move "was" to be before "detected" and delete revealed to make this sentence clearer.

(18) Figure 3: The resolution of the text in this figure is poor, making the type fuzzy and the subscripts in panel C unreadable.

(19) Section 3.3 generally: please include a brief statement somewhere in this section or even in the M&M  (where authors find most appropriate) of why and permanganate probing is used, the intent.  For those of us who are not well-schooled in studies of transcriptional, DNA binding, DNA structure, etc. but nonetheless interested in phage biology and gene expression this would be helpful.  I could find no such short explanation in the text.

(20) Line 323: Why "Surprisingly"?

(21) Lines 392-395: Why did you make this assumption? Please add a sentence to explain.

(22) Lines 423 and 425: Again confused by the use of "shall" here; perhaps "appears to be" instead?

(23) Line 429-430: Change "orchestrating" to "orchestrate."

(24) Line 461-462: How distant? Can provide a range of AA similarity?

(25) General question: Is the observation you made that 7-11 does not shut off host transcription via sigma 70 nor have any apparent middle promoters novel, or has this been observed before?  If novel, would not that up front in the abstract.

Author Response

Dear Reviewer 2, thank you for your comments. They were addressed in the revised manuscript and are detailed in the attached file. Our responses are in plain text. Your queries are italicized.

Sincerely,

Konstantin Severinov
